# GROUP-LEVEL BRAIN DECODING WITH DEEP LEARNING

## ABSTRACT

Decoding experimental variables from brain imaging data is gaining popularity, with applications in brain-computer interfaces and the study of neural representations. Decoding is typically subject-specific and does not generalise well over subjects. Here, we propose a method that uses subject embedding, analogous to word embedding in Natural Language Processing, to learn and exploit the structure in between subject variability as part of a decoding model, our adaptation of the WaveNet architecture for classification. We apply this to magnetoencephalography data, where 15 subjects viewed 118 different images, with 30 examples per image; to classify images using the entire 1s window following image presentation. We show that the combination of deep learning and subject embedding is crucial to closing the performance gap between subject- and group-level decoding models. Importantly, group models outperform subject models on low-accuracy subjects (but impair high-accuracy subjects) and can be helpful for initialising subject models. The potential of such group modelling is even higher with bigger datasets. To better enable physiological interpretation at the group level we demonstrate the use of permutation feature importance developing insights into the spatio-temporal and spectral information encoded in the models. All code is available on GitHub[1].

## 1 INTRODUCTION

In recent years, decoding has gained in popularity in neuroscience (Kay et al., 2008), specifically decoding external variables (e.g. stimulus category) from internal states (i.e. brain activity). Such analyses can be useful for brain-computer interface (BCI) applications (Willett et al., 2021) or to gain neuroscientific insights (Guggenmos et al., 2018; Kay et al., 2008). Analysing deep learning methods on such data is also beneficial for the machine learning community. Namely, the small, noisy, high-dimensional datasets test the limits of popular architectures on real data and demand research into new methods (Zubarev et al., 2019; Kostas et al., 2021). Applications of decoding to brain recordings typically fit separate (often linear) models per dataset, per subject (Guggenmos et al., 2018; Dash et al., 2020b). This has the benefit that the decoding is tuned to the dataset/subject, but has the drawback that it is unable to leverage knowledge that could be transferred across datasets/subjects. This is especially desirable for the field of neuroimaging, because gathering more data is expensive and often impossible (e.g. in clinical populations). More practical drawbacks of subject-specific (subject-level) models include increased computational load, a higher chance of overfitting, and the inability to adapt to new subjects. We aim to leverage data from multiple subjects and train a shared model that can generalise across subjects (group-level). A conceptual visualisation of subject-level (SL) and group-level (GL) models is given in Figure 1.

Magnetoencephalography (MEG) measures magnetic fields induced by electrical activity in the brain, and it is one of the main noninvasive brain recording methodologies, next to electroencephalography (EEG) and functional Magnetic Resonance Imaging (fMRI). Due to high temporal resolution and relatively good spatial resolution, MEG is an excellent method for studying the fast dynamics of brain activity. MEG is highly suitable for decoding analyses (Du et al., 2019), which is mostly done using SL models. This is because between-subject variability of neuroimaging data limits the application of a single shared model between subjects without capturing the structure of between-subject variability (Olivetti et al., 2014; Li et al., 2021). Such an approach, which we call naive group modelling,

---

[1]Anonymized.

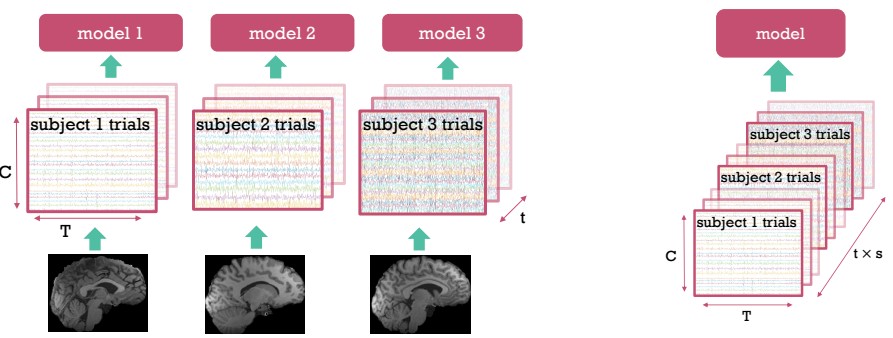

(a) Subject-level (SL) models.        (b) Naive group-level (GL) model.

Figure 1: Comparison of subject-level (SL) and naive group-level (GL) modelling. (a) A separate model is trained on the trials (examples) of each subject. (b) A single, shared model is trained on the trials of all subjects without capturing between subject variability. Each trial is **C** x **T** (channels x timesteps) dimensional. Each of the **s** subjects has **t** trials.

effectively pretends that all data comes from the same subject (see Figure 1b). Between-subject variability has multiple sources, such as different anatomical structures, different positions in the scanner, signal-to-noise ratio, etc. (Saha & Baumert, 2020). To overcome this, we propose a general architecture capable of jointly decoding multiple subjects with the help of subject embeddings (Figure 2). The scope of this paper is full-epoch decoding, and comparisons with sliding-window decoding approaches often used in neuroscience are left for future work.

To qualify how we aim to improve on SL models, we will next describe the two main approaches to evaluating decoding models, with different underlying assumptions and goals. One approach is to construct separate train and test splits for each subject that are made up of different, non-overlapping trials. This can be called *within-subject splitting evaluation*. SL models are evaluated by definition in this way, and it is a very common setup in the neuroscience literature (Guggenmos et al., 2018; Cooney et al., 2019b; Cichy & Pantazis, 2017; Dash et al., 2020b;a; Nath et al., 2020). In this work, our main aim is to improve over SL models in the context of within-subject splitting evaluation and improve the prediction of left-out trials, by using a single group decoding model that generalises across subjects. We call this GL method *across-subject* decoding. We are motivated by the fact that GL models that perform well in this manner can be useful for gaining neuroscientific insights that are relevant at the group level, as we will show in Sections 4.4 and 4.5. The other prominent approach to evaluating group models, leave-one-subject-out (LOSO) analysis, is also presented in Section 4.3. In this scenario, GL models are trained on data from multiple subjects and tested on a new, unseen subject (Zubarev et al., 2019), which can be especially useful in zero-shot BCI applications. Although in this case, we find no improvement using our embedding-aided group model, we think this may change with larger datasets with many more subjects. Our aim is to improve across-subject decoding of MEG data by using a group model that generalizes across subjects. To be clear this objective and the datasets we use are not related to any kind of direct BCI application.

We make the following contributions using a MEG dataset with visual task (Cichy et al., 2016): 1. A GL model with subject embeddings is introduced, substantially improving over naive group modelling. 2. Insight is provided into how non-linearity and subject embedding helps group modelling. 3. Neuroscientific insights are gained from the deep learning-based decoding model. 4. Analysis of model weights reveals how meaningful spatio-temporal and spectral information is encoded.

## 2 RELATED WORK

Decoding can be applied to most tasks/modalities, such as images (Cichy et al., 2016), phonemes (Mugler et al., 2014), words (Cooney et al., 2019b; Hultén et al., 2021), sentences (Dash et al., 2020b), and motor movements such as imagined handwriting (Willett et al., 2021), jaw movements (Dash et al., 2020a), or finger movements (Elango et al., 2017). Here, we used image categorisation because it is a widely studied decoding task and we had access to a dataset which is relatively large for the field of neuroimaging. Our results should readily generalise to other decoding modalities. Chaibub Neto

et al. (2019) strongly argue against using SL modelling, especially for clinical applications. Recently, some transfer learning approaches have been proposed to deal with the problem of variability between subjects. Kostas & Rudzicz (2020) have proposed two distinct methods. The euclidean alignment is very similar to a spatial whitening of the data. We have tried this in conjunction with our group model, however, we found this to lower performance, and thus opted for a simpler channel-wise standardization. The other method, mixup regularization is entirely complementary to our approach and can be used in conjunction. This is a general regularization/data augmentation technique and does not specifically deal with inter-subject variability. Most transfer learning frameworks consist of applying a model trained on one subject to a different (target) subject (Elango et al., 2017; Dash et al., 2019; Cooney et al., 2019a; Olivetti et al., 2014; Halme & Parkkonen, 2018; Li et al., 2021). Some approaches use learnable affine transformations between subjects (Elango et al., 2017), while others finetune the whole model on target subjects (Cooney et al., 2019a; Dash et al., 2019). However, these approaches offer only a marginal improvement over naive group modelling and do not use a shared model across subjects. We aim to significantly improve this by using a general framework and model capable of decoding multiple subjects at once.

We must note that while there is a clear distinction between within-subject splitting and LOSO (leave-one-subject-out) evaluation, most of the transfer learning works that aim to improve in a LOSO scenario do use some form of training on the left-out subject (Zubarev et al., 2019). Thus, the distinction between the two evaluation frameworks gets less transparent and we discuss in Section 4.3 how our methods could be useful in this less stringent LOSO definition. We believe that ultimately the way a method is intended to be used in practise should determine how that method is benchmarked.

Transfer learning is popular in the wider machine learning field. Parallels can be drawn with domain adaptation (Long et al., 2015), or transferring knowledge from large to small datasets within the same domain (Wang et al., 2019; Zhuang et al., 2020). Natural language processing (NLP) datasets often contain data from widely different sources (Radford et al., 2022), but due to the sheer size of the dataset and model complexity, training on joint data achieves good results (Brown et al., 2020; Devlin et al., 2019). As discussed before, this naive concatenation of subjects does not work well on small neuroimaging datasets. Perhaps the most relevant parallels can be drawn with dialogue and speech modelling work, where inter-speaker differences are modelled using speaker embeddings (Li et al., 2016; Zhang et al., 2018; Saito et al., 2019; Mridha et al., 2021). Chehab et al. (2021) have similarly found that subject embeddings provide a small but significant improvement in encoding MEG data from a language task. However, limited information is provided on how subject embedding helps, and their results cannot be directly generalised to MEG *decoding*. We build on this work, expanding it to the task of decoding MEG data, and provide additional insight into how deep learning and subject embeddings help GL decoding models.

## 3 METHODS

### 3.1 DATA

In this work, a task-MEG dataset is used where 15 subjects view 118 different images, with each image viewed 30 times (Cichy et al., 2016). The data is publicly available[2], however, we obtained the continuous raw MEG data directly from the authors to be able to run our preprocessing pipeline using MNE-Python (Gramfort et al., 2013). Raw data is bandpass filtered between 0.1 and 125 Hz and line noise is removed with notch filters. After downsampling to 250 Hz, 1.024-second epochs are extracted, starting 100 ms before stimulus presentation. This resulted in 306 x 256-dimensional trials (channels x timesteps) from the 306 MEG sensors. Whitening is used to remove covariance between channels for SL models, whereas for GL models, a standardisation is performed per channel. We do multiclass decoding, predicting a separate probability for each of the 118 classes (images).

### 3.2 MODELS

Our deep learning model, the WaveNet Classifier, is inspired by previous approaches to applying WaveNet (van den Oord et al., 2016) for classification (Zhang et al., 2020). We do not aim to design a new kind of architecture for decoding MEG data, but rather build our model based on previously

---

[2]http://userpage.fu-berlin.de/rmcichy/fusion_project_page/main.html

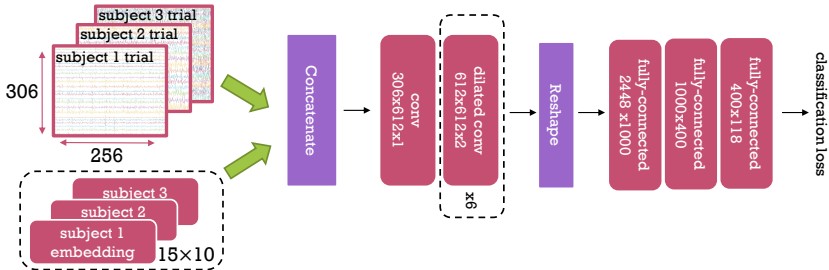

Figure 2: Group-level WaveNet Classifier with subject embeddings. Dashed boxes represent parts of the model which differ between SL and GL versions of our architecture. Red boxes represent learnable parameters. For convolutional layers the numbers represent *input channels* x *output channels* x *kernel size*. For fully-connected layers the numbers represent *input neurons* x *output neurons*. The embedding layer dimensionality is given as **s** x **E** (15 x 10), where **s** is number of subjects, and **E** is the embedding size. Embeddings are concatenated with input trials to provide information about which trial is coming from which subject. The classification loss is cross-entropy.

proven CNN-based architectures for EEG and MEG data (Lawhern et al., 2018). While Wavenet has proved to be successful in the audio domain, to the best of our knowledge, there is no prior work applying Wavenet to neural decoding. Our model consists of 2 parts: the (temporal) convolutional block, intended to act as a feature extractor; and the fully-connected block, which is designed for classification (Figure 2). The convolutional block uses a stack of 1D dilated convolutional layers, which include dropout and the inverse hyperbolic sine activation function. The dilated convolutions in WaveNet are effective for modelling time series data as successive layers extracts complementary frequency content of the input (Borovykh et al., 2018). Since the dilation factor is doubled in successive layers, the receptive field of the convolutional block is $2^{num\_layers}$. Given there is no pooling and a convolution stride of 1, the output of each layer preserves the temporal dimensionality[3]. At the end of the convolutional block, we downsample temporally by the size of the receptive field. In the model with 6 convolutional layers, this means that the initial input of size 256 is downsampled by a factor of 64, resulting in 4 values per channel. Next, this output is flattened and fed into a fully-connected block. The final output is a logit vector corresponding to the 118 classes. The model is trained with the cross-entropy loss for classification.

For SL modelling, the Wavenet Classifier contains 3 convolutional layers, whereas for group modelling it has 6, further motivated in Section 4. Our improvement of the naive group model includes subject embeddings, which are introduced as a way of dealing with between-subject variability, similarly to Chehab et al. (2021). Like word embeddings in NLP, each subject has a corresponding dense vector (Mikolov et al., 2013). This vector is concatenated with the channel dimension of the input trial across all timesteps. Thus, in the embedding-aided GL model, input trials are (**C+E**) x **T** dimensional, where **E** is the embedding size. Subject embeddings are learnt together with other model weights using backpropagation. We reasoned that an embedding-aided model can learn general features across subjects, with the capability of adapting its internal representations for each subject.

### 3.3 MODEL ANALYSIS

In this section, we describe several approaches to uncover the information encoded in the WaveNet Classifier. In *Kernel FIR Analysis*, we investigate the frequency characteristics of the convolutional kernels. Random noise is fed into a trained model, and the power spectral density of the output of specific kernels is computed to assess their finite impulse response (FIR) properties. Permutation Feature Importance (PFI) is a powerful method to assess which features contribute the most to model performance (Altmann et al., 2010; Chehab et al., 2021). Chehab et al. (2021) specifically have showed the power of PFI for analysing how certain language features like word frequency affect the performance of forecasting MEG data at different temporal and spatial locations. Conversely, in MEG decoding we can directly assess both spatial and temporal information by permuting across timesteps (for each channel) and across channels (for each timestep), respectively. We call these

---

[3]Except the amount that gets chopped off because of the kernel size itself, since we do not use padding.

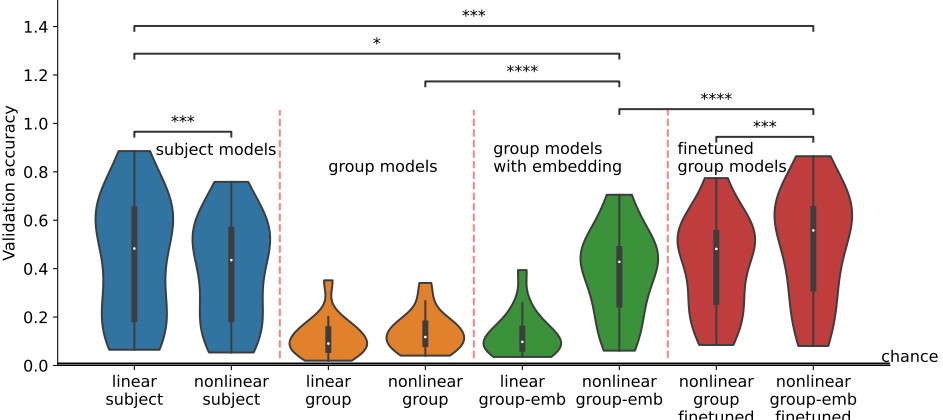

Figure 3: SL and GL models evaluated on the validation set of each subject. Paired samples T-tests are shown for comparisons of interest ($* = p < 5e-2, ** = p < 1e-2, *** = p < 1e-3, **** = p < 1e-4$). The `non-linear group-emb finetuned` model is finetuned separately on each subject, initialized with the `non-linear group-emb` model. Chance level is $1/118$.

temporal and spatial PFI, respectively. A decrease from the original accuracy (with unpermuted inputs) indicates that (visual) stimulus-related information is present in the MEG data in certain time periods or sensors. Alternatively, when looking at individual kernels, our feature importance measure is the absolute difference between the kernel output using the original and permuted inputs. We reason that a more important feature will cause a higher output deviation. To assess the frequency sensitivity of individual kernels, we introduce the *spectral PFI*. First, the data in each channel of each trial is Fourier transformed, and the Fourier coefficients are shuffled across channels for each frequency (or frequency band). Then, the inverse Fourier transform is computed, obtaining a trial with disrupted information in specific frequency bands.

## 4 RESULTS

### 4.1 SUBJECT EMBEDDING CLOSES THE GAP BETWEEN SL AND GL MODELS

Our main evaluation metric is classification accuracy across the 118 classes. Train and validation splits with a 4:1 ratio were constructed for each subject and class. This means that classes are balanced (i.e., contain the same amount of examples) across subjects and splits. The first 20% of the continuous MEG data is used to extract validation examples, with the remaining used for training trials. SL and GL models are trained and evaluated on the same splits. For each model, an extra training is conducted wherein the (linear) identity function is used as an activation function to assess the influence of non-linearity, which is the bedrock of deep learning. Linear and non-linear models are trained for 500 and 2000 epochs (full passes of the training data), respectively, with the Adam optimiser (Kingma & Ba, 2015). More training details are provided in the Appendix. We compute paired samples T-tests for comparisons of interest over methods, where the pairing is within subject and samples are the subject-level mean accuracies over validation trials. We used PyTorch for training (Paszke et al., 2019) and several other packages for analysis and visualisation (Pedregosa et al., 2011; Virtanen et al., 2020; Harris et al., 2020; Wes McKinney, 2010; Waskom, 2021; Hunter, 2007).

Validation accuracies for all models are shown in Figure 3. Interestingly, at the subject level, linear models performed slightly better than non-linear (4% increase, $p = 5.7e-4$). We think that both the limit in data size and noise levels in the data contribute to the subpar performance of non-linear models. Although some studies have found deep learning to improve over simpler linear models, this improvement is often marginal (Cooney et al., 2019b; Schirrmeister et al., 2017). Such results are difficult to generalise across different MEG datasets, due to variability in tasks, the number of subjects, and the amount and quality of data (Schirrmeister et al., 2017). The large between-subject variability common to MEG datasets is apparent, with individual subjects' accuracy ranging from 5% to 88%. As

expected, the naive group models, i.e. a naive application of either the linear or non-linear WaveNet Classifier to the group modelling problem, result in much worse performance than SL models (30% decrease). Inferring such high variability between so few subjects implicitly is not trivial. Adding subject embeddings to the non-linear model improves performance by 24% ($p = 1.9e − 6$), with no increase for the linear model. This shows that leveraging subject embeddings in conjunction with non-linear activations can narrow the gap with SL models (6% difference, $p = 1.3e − 2$). Limiting the non-linearity to the first layer resulted in a subpar performance, similar to that of a linear model. This indicates that non-linearity is needed within multiple layers to benefit from subject embeddings.

We also finetuned the embedding-aided GL model on the training data of each subject separately (`non-linear group-emb finetuned`) for 500 epochs. We effectively use the GL model as an initialisation for SL models, improving over SL models trained from scratch, achieving 50% accuracy (5% increase, $p = 1e − 3$). This shows that representations learned at the group level are useful for SL modelling. In contrast, finetuning a naive group model (`non-linear group finetuned`) only achieved 42% accuracy (8% lower), showing that finetuning performance depends on good GL performance. Thus, in addition to closing the gap between SL and GL modelling, finetuning our embedding-aided model provides the best overall accuracy for SL modelling. The variance of `non-linear group-emb` (0.19) and `non-linear group-emb finetuned` (0.24) is lower than the SL models (0.26). Generally, the group model is reducing between-subject variability. In the case of `non-linear group-emb`, 4 subjects with generally low accuracies (15-30%) had higher accuracies than SL models (even though the mean across subjects is lower). This shows that training a group model is a good approach to improve low-accuracy subjects. Indeed, strong negative correlations of -0.88 and -0.54 are obtained between the SL accuracy and the change in accuracy achieved by the `non-linear group-emb` and `non-linear group-emb finetuned` models, respectively. Comparing finetuning to from-scratch SL models, only 2 high-accuracy subjects are slightly worse, and generally low/mid-accuracy subjects show more improvement than high-accuracy subjects (see Appendix for plots). In addition to the main dataset used in this section we also analysed our main findings on another publicly available visual MEG dataset (see Appendix for results).

In summary, these results suggest the following recommendations for decoding MEG task data. 1. Subject embeddings and non-linearity should be used for achieving good group models. 2. GL models can be used to improve over SL models on low-performance subjects. 3. For the best SL performance, the finetuning approach should be used, benefitting low-performance subjects the most.

## 4.2 INSIGHTS INTO SUBJECT EMBEDDINGS AND OTHER MODELLING CHOICES

For the embedding-aided GL setup, 4 further models were trained for 5-fold cross-validation. Average accuracy was 37.4% (as opposed to the 38% reported in Figure 3), with a 95% confidence interval of 0.8%. Thus, the proposed GL model is robust to different random seeds and dataset partitions. A more extensive robustness analysis is omitted due to computational constraints. In non-linear SL models, performance improves as we use fewer convolutional layers. This effect plateaus at 3 layers, whereas for non-linear GL models using more layers improved accuracy (see Table 1 in the Appendix for exact accuracies). Thus, SL models seem to rely more on the fully-connected block as they are unable to extract good features, and GL models rely more on the convolutional block to learn shared features across subjects. To be clear, because of how we perform the downsampling after the convolutional layers (described in Section 4.1), using fewer convolutional layers increases the overall parameter count because the fully-connected block has to be enlarged. Thus, the group model (with 6 conv layers), is about 2.5x smaller than the SL models (with 3 conv layers). However, `non-linear group-emb finetuned` models achieve higher accuracy than from-scratch SL models. This shows that when initialised well (with a group model trained on multiple subjects) even SL models can benefit from non-linearity and more convolutional layers.

We tried different approaches to understand how subject embeddings help the group model. A clustering or 2D projection of the embedding space such as PCA or t-SNE (Van der Maaten & Hinton, 2008) did not show any clusters (see Figure 8 in the Appendix). This is likely to be a consequence of only having 15 subjects, since cases where such visualisations work well (Liu et al., 2017) typically have thousands of dimensions (e.g. words in word-embeddings). To assess whether the embeddings simply encode which subjects are good, we transformed the embeddings with PCA and correlated all components with the accuracies across subjects. We found no significant correlations; therefore,

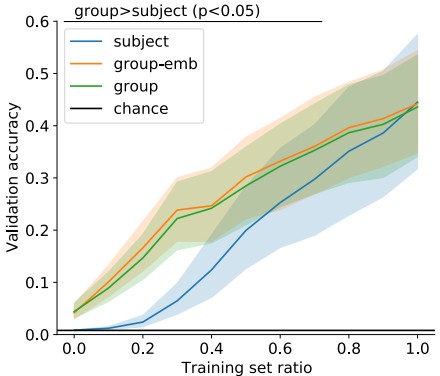

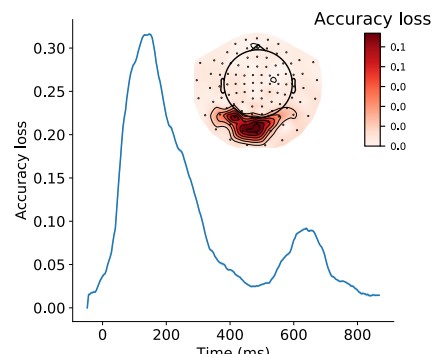

(a) Generalisation and finetuning on left-out subjects. The horizontal axis shows the amount of training data used from the left-out subject; a training set ratio of 0 corresponds to a zero-shot approach. `subject` is trained from scratch, while `group-emb` and `group` are initialised with the non-linear GL model with and without embeddings, respectively. The 95% confidence interval of the accuracy across left-out subjects is shown with shading.

(b) Temporal (line) and spatial (sensor space map) PFI for the `non-linear group-emb` model. For temporal PFI accuracy loss (vertical axis) is plotted with respect to time since visual image presentation (horizontal axis). Shading shows the 95% confidence interval which is not visible due to low variability. For spatial PFI, darker red shading is equivalent to higher accuracy loss.

Figure 4: Generalisation performance (a); Temporal and Spatial PFI (b).

embeddings do not appear to encode information about SL accuracy. To assess how much embeddings contribute to a trained model, we tried both setting the embeddings to zero and shuffling them. This reduced the accuracy to 10% (from 38%). Thus, embeddings encode crucial information to aid decoding, but the model is still better than chance without them. For further insights see Appendix.

### 4.3 LEAVE-ONE-SUBJECT-OUT EVALUATION

In this section we report leave-one-subject-out (LOSO) cross-validation results, a common evaluation setup with several use cases as described in Section 1. We also analyse how performance improves when we allow models to use increasing amounts of data (finetuning) from the left-out subject. We compared the naive group and group-embedding models (non-linear) with a from-scratch SL model (linear) trained only on the left-out subject. To be clear, as in the finetuning approach in Section 4.1 the group models are only used as initialisation for an SL model trained on the left-out subject. The only difference is that the subject on which we are finetuning is not in the training data of the group models (as opposed to the finetuning setup in Section 4.1). For the embedding-aided group model, the left-out subject's embedding was initialised randomly. In the LOSO (zero-shot) evaluation, group models achieve 5% accuracy (Figure 4a). Up to the case when 70% of the training set is used a finetuned GL model is much better than a from-scratch SL model ($p < 0.05$, corrected for multiple comparisons). The benefit of GL models in LOSO analysis has been previously established (Elango et al., 2017). Unsurprisingly, the embedding-aided group model does not improve over the naive group model, but is, importantly, not worse. As opposed to the finetuning setup in Figure 3, when adapting to new subjects, better group performance does not translate to better finetuning performance, as the model is unable to leverage the new subject's embedding.

Exploiting subject embeddings in a pure LOSO framework is not trivial, as some additional approach is needed to initialise/learn the embedding of the left-out subject in an unbiased manner. In larger datasets with more subjects, between-subject similarities in the embeddings could be exploited and different heuristics explored, e.g. initialising the embedding with the average of all learned subject embeddings. However, research aimed at improving performance in new subjects often leverages transfer learning in some way, where a limited amount of data from the new subject can be used (Zubarev et al., 2019). In this scenario, we think our across-subject group model could be helpful, by, for example, using the limited data from the new subject or by learning a useful embedding for the new subject in an unsupervised manner. As we have shown in Section 4.1 this could be especially useful for subjects with low performance. As opposed to a naive continuation of the trends

in Figure 4a, we expect that with more trials, the gap between group initialisation and training from scratch would continue, up to some limit. We believe that the reason why the gap closes at 100% training data is due to the ratio of training and validation sets and the low number of examples. The small validation set (6 examples per class) is probably not representative of the full data distribution.

## 4.4 Neuroscientific insights are gained from a deep learning based GL model

An established critique of deep learning models applied to neuroimaging data is the lack of interpretable insight (Murdoch et al., 2019) they provide about the underlying neural processes that drive the decoding. To gain such neuroscientific insights, it is useful to assess the time- and space-resolved information/discriminability within trials. As mentioned in Section 3.3, permutation feature importance (PFI) is a suitable, model-agnostic measure to do this. Figure 4b shows the temporal and spatial PFI of the non-linear group-embedding model. To make the results robust and smooth, the shuffling for temporal PFI was applied to 100 ms windows, and magnetometers and gradiometers in the same location were shuffled together for spatial PFI. Time windows or channels with higher accuracy loss than others are interpreted as containing more information about the neural discriminability of the visual images. This indicates when and where information processing related to the presented images is happening in the brain. Temporal PFI shows a large peak around 150 ms which is often observed in neuroscientific decoding studies employing sliding window analysis to assess temporal information content (Higgins et al., 2022). After this, the information content rapidly decreases, with a second, smaller peak around 650 ms, which could correspond to a brain response following the end of image presentation at 500 ms. Spatial PFI shows that the most important channels are in the back of the head in the sensors in visual areas, which is as expected for a visual task. We found good agreement between this PFI analysis and a standard gradient-based analysis, further detailed in the Appendix.

## 4.5 Weights encode meaningful spatio-temporal and spectral information

To provide further insight into our `non-linear group-emb` model, we next show that neuroscientifically interpretable spatial, temporal, and spectral information can be obtained by analysing the learnt weights. All visualisations are from 3 convolutional layers, with all 6 layers shown in the Appendix. Kernels within a layer seem to have similar temporal sensitivity, even though only 5 are shown from over 1e5 total kernels (Figure 5c). Output deviations are standardised to compare temporal PFI across kernels with different output magnitudes. In the early layers, sensitivity peaks around 100 ms (as in Figure 4b), then rapidly decreases, eventually climbing again slowly. Kernels in early layers have somewhat random spatial sensitivity (Figure 5a), but this gets narrowed down to channels over the visual cortex in deeper layers, with some differences between individual kernels. This sensitivity is similar to the spatial features that were shown to be most informative for classification performance (see Figure 4b). Figure 5b shows the temporal profile of the spatial PFI. This is achieved by focusing the shuffling to 100 ms time windows and 4-channel neighbourhoods (3 closest channels for each channel), repeated across all timesteps and channels. Spatial sensitivity does not seem to change with time; i.e. the most important channels are always the same.

Spectral PFI measures the change in kernel output to perturbations in specific frequency bands (see Figure 10a in Appendix). Across all layers and kernels, the profile has a 1/f (frequency) shape with a clear peak at 10 Hz. These are common features of the MEG signal (Demanuele et al., 2007; Drewes et al., 2022), indicating that the spectral sensitivity of the kernels coincides with the power spectra of the data. We also looked at the spectral PFI of 4-channel neighbourhoods (Figure 10b in Appendix) and found that kernels are sensitive to the same channels (in the visual area) across all frequencies, with these channels having larger 10 Hz peaks. Kernel FIR analysis shows the power spectra of kernels' outputs when input examples are Gaussian noise (see Figure 10c in Appnedix). The subject embedding was set to a subject with average accuracy. The power spectra were normalised to make visual comparisons between kernels easier. Since the WaveNet architecture uses dilated filters with only 2 values per filter, early layers show broad filtering characteristics, but already in layer 2 more emphasis is put on lower frequencies. In deeper layers, filters (kernels) become more tuned to specific frequencies, generally below 20Hz. This is in line with the spectral properties of MEG data as discussed above. Both the spectral PFI and kernel FIR analysis shows that there is significant variability between the spectral information encoded by various kernels. From the analysis presented in this section, we can conclude that kernels are sensitive to interpretable temporal, spatial, and spectral features of the MEG data.

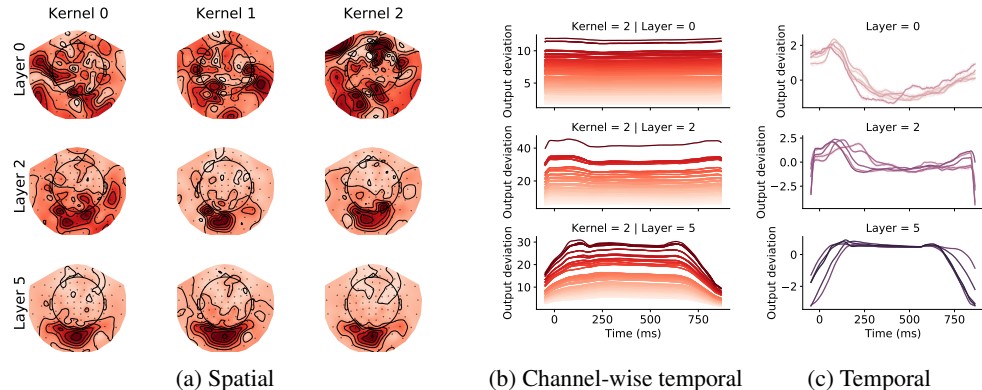

(a) Spatial       (b) Channel-wise temporal       (c) Temporal

Figure 5: Spatio-temporal insights can be obtained using PFI. Spatial (a), channel-wise temporal (b), and temporal (c) PFI across `non-linear group-emb` kernels within 3 layers (rows). For spatial PFI, kernels are plotted separately; whereas for temporal PFI, 5 kernels (lines) are plotted together. Channel-wise temporal PFI shows the temporal PFI of each channel for Kernel 2. Channel colouring is matched to the corresponding spatial PFI map, and darker reds mean higher output deviation. For temporal PFI, output deviation is normalised. The horizontal axis shows the time elapsed since image presentation, for both temporal PFI types. 95% confidence intervals are shown with shading.

## 5   DISCUSSION

We have proposed a deep learning-based group-level (GL) model that improves significantly over naive GL modelling, in which nothing is learnt about the nature of between-subject variability, achieving similar performance to subject-level (SL) models. We have shown how subject embeddings and non-linearity are crucial for this. These are important insights towards the goal of using group models in decoding neuroimaging data, which would allow for better use of this inherently limited resource. Our proposed GL model is especially beneficial in the case of low-accuracy subjects, and can be useful as an initialisation for SL models in the case of across-subject decoding. In addition, we have demonstrated that we can use PFI to obtain insight into which time points and channels contributed to the decoding and to obtain meaningful information encoded in convolutional kernels. Using this and other methods, such as representational similarity analysis, neuroscientific investigations can be performed at the group level.

We found that in the context of leave-one-subject-out evaluation, using subject embeddings did not improve performance. The outcome may be different when our method is applied to larger neuroimaging datasets with more subjects or when subject embeddings are combined with other transfer learning methods, as discussed in Section 4.3. Further research is needed into deep learning models capable of implicitly learning inter-subject variability. An important question is whether scaling up models on large datasets would achieve this goal. We expect the subject embedding and group modelling to generalise to different task and recording modalities (EEG, fMRI, etc.) because they face similar decoding challenges. The specific Wavenet-based model is readily generalisable to other electrophysiological data such as EEG and Electrocortiocography, because of the same temporal dynamics they capture. The scope of this work is full-epoch decoding, and it remains to be seen how our methods perform with sliding-window decoding.

Other than being useful for fine-tuning, our embedding-aided group model can be useful in the case of much larger datasets where we cannot afford to have a separate model for each subject. In neural decoding, group models are widely understood to perform worse than individual models ((Guggenmos et al., 2018; Dash et al., 2020b)). But why is this? By plotting per-subject performance in both kinds of models (see Figure 9 in the Appendix), we see something revealing. While group results are worse on average, some subjects are decoded better in the group model than in their individual models. These generally appear to be the subjects who perform worst on the individual models. This suggests that group models could be successfully used for some subjects if those subjects could be identified. Our results suggest follow-up studies to understand why some subjects performed better or worse.

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

## A  APPENDIX

Dropout was set to 0.4 and 0.7, and a batch size of 590 and 59 was used for GL and SL models, respectively. Learning rate was set to 0.0001 for GL, and 0.00005 for SL models. Training of a single SL and GL model took 5-15 minutes and 4 hours on an NVIDIA A100 GPU, respectively. For linear models, validation losses (cross-entropy) and accuracies were negatively correlated, i.e. loss decreases while accuracy increases, and eventually both suggested overfitting. Since non-linear models are more expressive, they overfitted sooner according to the loss, but accuracy kept improving until it reached a plateau, never overfitting. Analysing the loss distribution across validation examples (for non-linear models) shows that even during overfitting most examples' loss keeps decreasing with a few high-loss outliers disproportionately influencing the mean. Since accuracy is binary, outliers are diminished, explaining the apparent difference in learning behaviour. For linear models, this unintuitive behaviour was not observed probably due to inherent model simplicity.

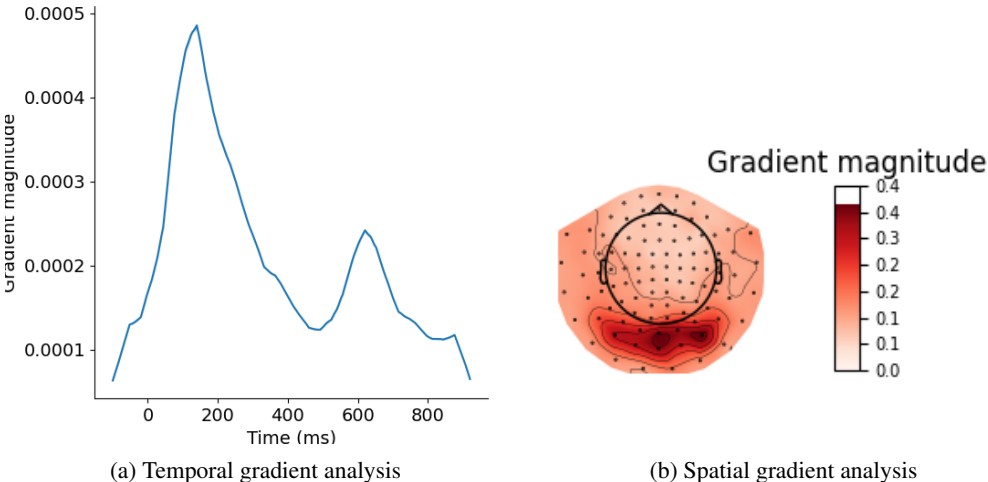

(a) Temporal gradient analysis        (b) Spatial gradient analysis

Figure 6: Using gradient analysis by backpropagating the loss to randomly initialized inputs. In (a) we can see the temporal profile of the gradients averaged over channels. In (b) we can see the spatial profile of the gradients averaged over time.

We analysed our main findings on another publicly available visual MEG dataset with 92 different images (15 subjects, and 30 trials per image) (Cichy et al., 2014). Linear SL models achieved 35% accuracy, whereas a linear group model without embeddings had 12%, and a nonlinear group model with embeddings had 28%. Thus we can see that our approach behaves similarly on this dataset, improving a lot over the naive group baseline, but not quite achieving the performance of the linear SL models. Finetuning the group model separately on individual subjects achieved 38% accuracy surpassing from-scratch SL models.

Training with an embedding dimensionality of 3 and 14, resulted in 20% and 38% accuracy, respectively. We tried these two settings to see how embedding size in the lower and upper limits influences performance. As an embedding dimensionality of 14 performs the same as 10, we could draw the conclusion that 10 is not a limiting factor. From the much worse result with an embedding dimensionality of 3 we could draw the conclusion that compressing the embedding representations too much is not possible. As with the clustering analysis, this is likely to be due to having few subjects. To gain further insight into the learned subject embeddings we computed accuracy on each subject's validation data using other subjects' embeddings. In the resulting subject-by-subject confusion matrix the value in the $i$-th row and $j$-th column shows how well the embedding of subject $i$ can be replaced with the embedding of subject $j$ (Figure 7). After division with the original accuracies the metric shows how much accuracy can be retained when swapping subject embeddings. Some subjects' embedding cannot be replaced by others (e.g. subject 3), and some subjects' embedding can be more easily replaced (e.g. subject 12). Conversely, some subjects' embeddings are more general as they can replace many others (e.g. subject 14), and some are less general (e.g. subject 2). We tried clustering this matrix, and looked at correlation with both embedding distance and subject accuracy, however no meaningful results were found.

We compared the accuracy-based PFI analysis with a standard gradient-based analysis where a saliency map is obtained by backpropagating to randomly initialized inputs (Figure 6. We smoothed the temporal profile with the same window size as for the PFI analysis. Temporally we can see that the agreement between the two methods is high, with peaks aligning very well (less than 10ms difference). Spatially the two methods do show some differences, but overall gradient analysis still points to the most important information being in the visual cortex.

|  | linear subject | nonlinear subject | nonlinear group-emb |
|---|---|---|---|
| **3 conv layers** | 0.45 | 0.39 | 0.22 |
| **6 conv layers** | 0.41 | 0.25 | 0.38 |

Table 1: Effect of number of convolutional layers on the validation accuracy of two subject-level and one group-level model.

|  | E0 | E1 | E2 | E3 | E4 | E5 | E6 | E7 | E8 | E9 | E10 | E11 | E12 | E13 | E14 |
|---|---|---|---|---|---|---|---|---|---|---|---|---|---|---|---|
| V0 |  | 0.16 | 0.09 | 0.08 | 0.26 | 0.21 | 0.13 | 0.12 | 0.13 | 0.18 | 0.12 | 0.10 | 0.20 | 0.11 | 0.19 |
| V1 | 0.13 |  | 0.04 | 0.11 | 0.11 | 0.09 | 0.10 | 0.13 | 0.11 | 0.18 | 0.12 | 0.10 | 0.13 | 0.03 | 0.15 |
| V2 | 0.09 | 0.05 |  | 0.14 | 0.06 | 0.10 | 0.08 | 0.10 | 0.10 | 0.11 | 0.13 | 0.10 | 0.10 | 0.08 | 0.14 |
| V3 | 0.04 | 0.04 | 0.07 |  | 0.03 | 0.06 | 0.03 | 0.05 | 0.05 | 0.06 | 0.06 | 0.06 | 0.03 | 0.03 | 0.07 |
| V4 | 0.36 | 0.16 | 0.12 | 0.07 |  | 0.17 | 0.33 | 0.24 | 0.13 | 0.25 | 0.16 | 0.16 | 0.23 | 0.15 | 0.35 |
| V5 | 0.15 | 0.08 | 0.08 | 0.11 | 0.11 |  | 0.09 | 0.16 | 0.18 | 0.14 | 0.08 | 0.08 | 0.27 | 0.11 | 0.11 |
| V6 | 0.14 | 0.11 | 0.05 | 0.05 | 0.23 | 0.08 |  | 0.09 | 0.09 | 0.15 | 0.08 | 0.12 | 0.09 | 0.10 | 0.28 |
| V7 | 0.23 | 0.16 | 0.17 | 0.13 | 0.26 | 0.29 | 0.21 |  | 0.24 | 0.21 | 0.26 | 0.33 | 0.25 | 0.23 | 0.29 |
| V8 | 0.15 | 0.10 | 0.05 | 0.07 | 0.11 | 0.11 | 0.07 | 0.17 |  | 0.13 | 0.14 | 0.13 | 0.18 | 0.07 | 0.10 |
| V9 | 0.25 | 0.24 | 0.12 | 0.12 | 0.19 | 0.15 | 0.19 | 0.20 | 0.16 |  | 0.27 | 0.16 | 0.20 | 0.03 | 0.32 |
| V10 | 0.29 | 0.13 | 0.11 | 0.10 | 0.16 | 0.15 | 0.15 | 0.24 | 0.24 | 0.25 |  | 0.15 | 0.25 | 0.06 | 0.23 |
| V11 | 0.11 | 0.15 | 0.05 | 0.10 | 0.18 | 0.12 | 0.10 | 0.21 | 0.13 | 0.19 | 0.10 |  | 0.15 | 0.18 | 0.18 |
| V12 | 0.60 | 0.30 | 0.23 | 0.19 | 0.40 | 0.49 | 0.30 | 0.37 | 0.47 | 0.42 | 0.49 | 0.42 |  | 0.28 | 0.28 |
| V13 | 0.10 | 0.06 | 0.05 | 0.05 | 0.14 | 0.07 | 0.10 | 0.10 | 0.07 | 0.04 | 0.06 | 0.14 | 0.13 |  | 0.09 |
| V14 | 0.19 | 0.10 | 0.05 | 0.06 | 0.30 | 0.10 | 0.29 | 0.15 | 0.10 | 0.20 | 0.12 | 0.17 | 0.18 | 0.17 |  |

Figure 7: Subject embedding confusion matrix. Columns (E0-E14) refer to subject embedding indices and rows (V0-V14) refer to subject validation sets. Greener shading (higher values) shows subjects with higher retained accuracy when their embeddings are swapped.

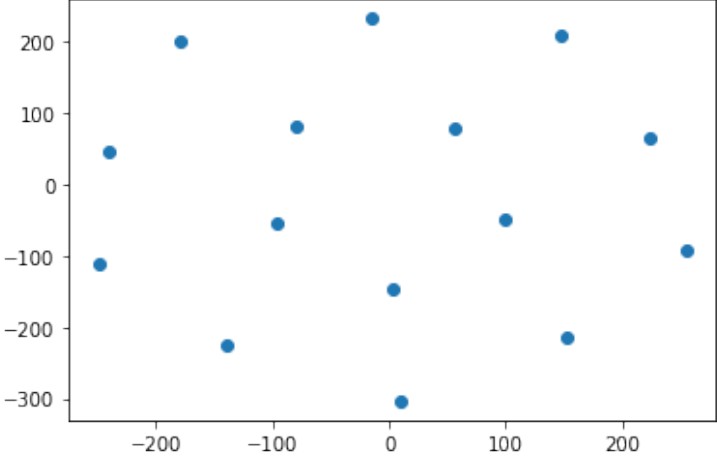

Figure 8: 2D Tsne projection of the subject embeddings in the nonlinear group-embedding model.

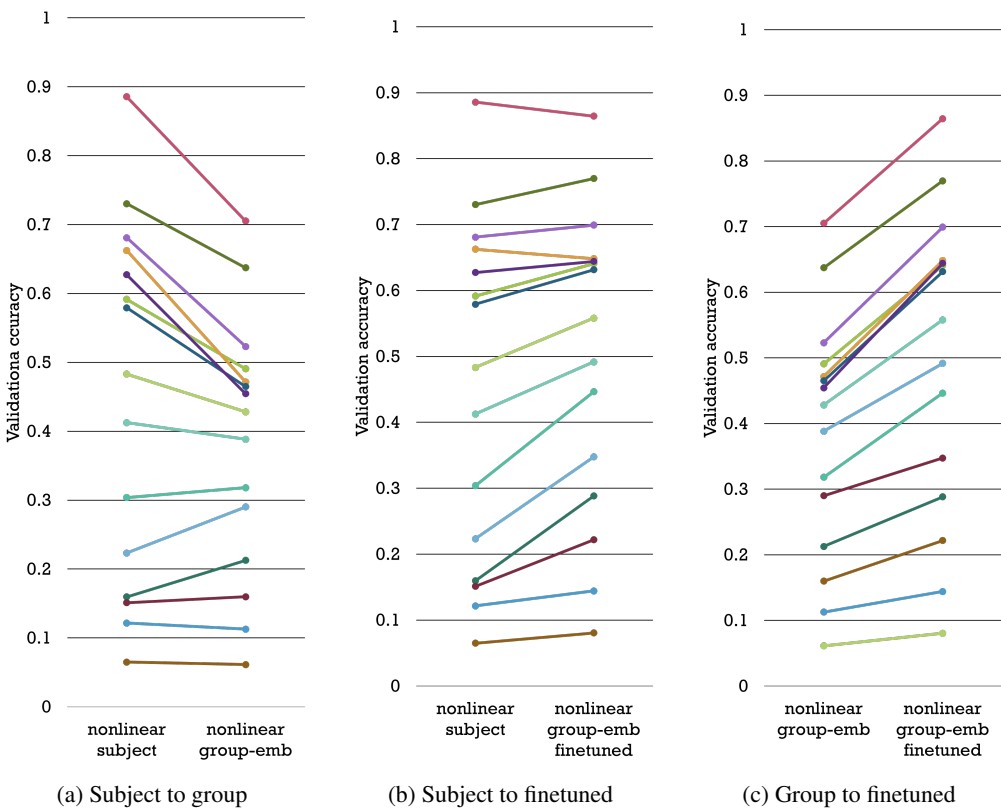

(a) Subject to group     (b) Subject to finetuned     (c) Group to finetuned

Figure 9: Accuracy changes across all 15 subjects (individual colours), when comparing SL, GL, and finetuned group models. Group models clearly reduce the variability of accuracies across subjects, and are especially helpful for low-accuracy subjects. When finetuning our group model on individual subjects (c), we can see that accuracy increases for all subjects, and especially for high accuracy subjects. This is unsurprising because these subjects have good enough data on their own for SL models to be able to learn well. As seen in (a) and (b) these high-accuracy subjects are usually impaired by GL models, exactly for the aforementioned reason.

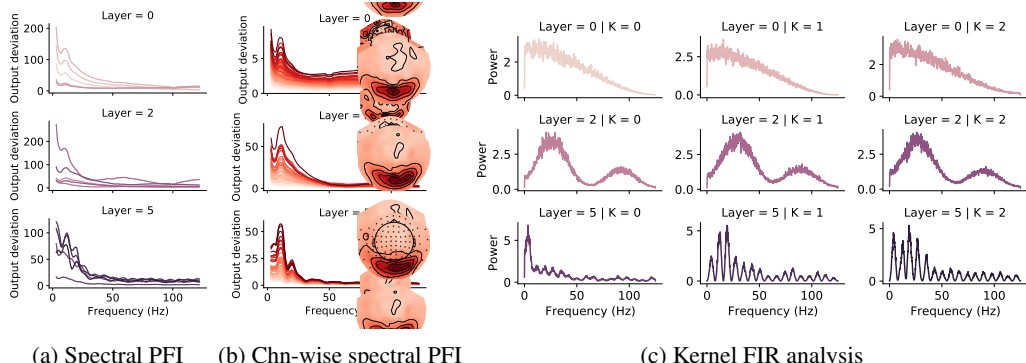

(a) Spectral PFI     (b) Chn-wise spectral PFI     (c) Kernel FIR analysis

Figure 10: Frequency sensitivity of kernels via spectral PFI (a), channel-wise spectral PFI (b), and frequency characteristics via kernel FIR analysis (c), from 3 layers (rows). Kernels are plotted together (lines) for spectral PFI, and in separate columns for kernel FIR analysis (normalised). Each channel-wise spectral PFI plot is for 1 kernel, where lines show the spectral PFI of corresponding channels in the sensor space map. 95% confidence intervals are shown with shading for spectral PFI. Due to small variability across permutations, this is barely visible. For spectral PFI the band-width was set to 5 Hz to obtain a smooth frequency profile.

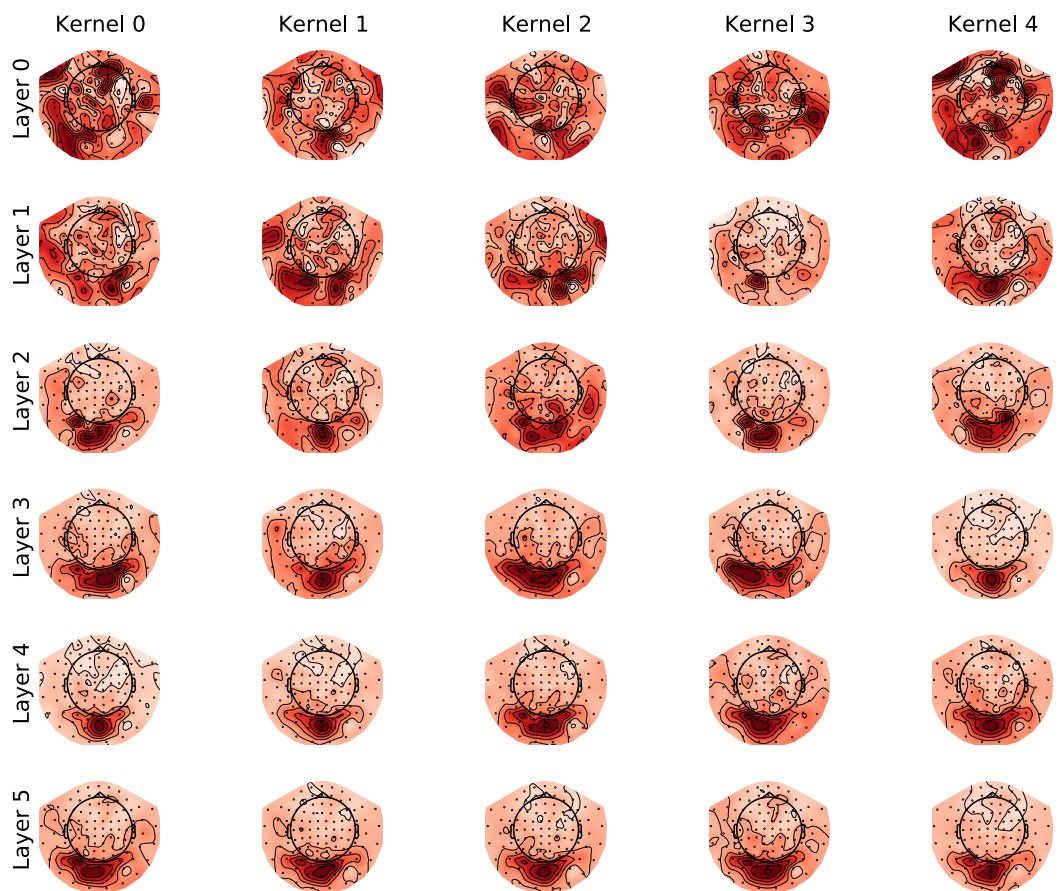

Figure 11: Spatial PFI across 6 layers (rows), with 5 kernels per row. Darker reds mean higher output deviation.

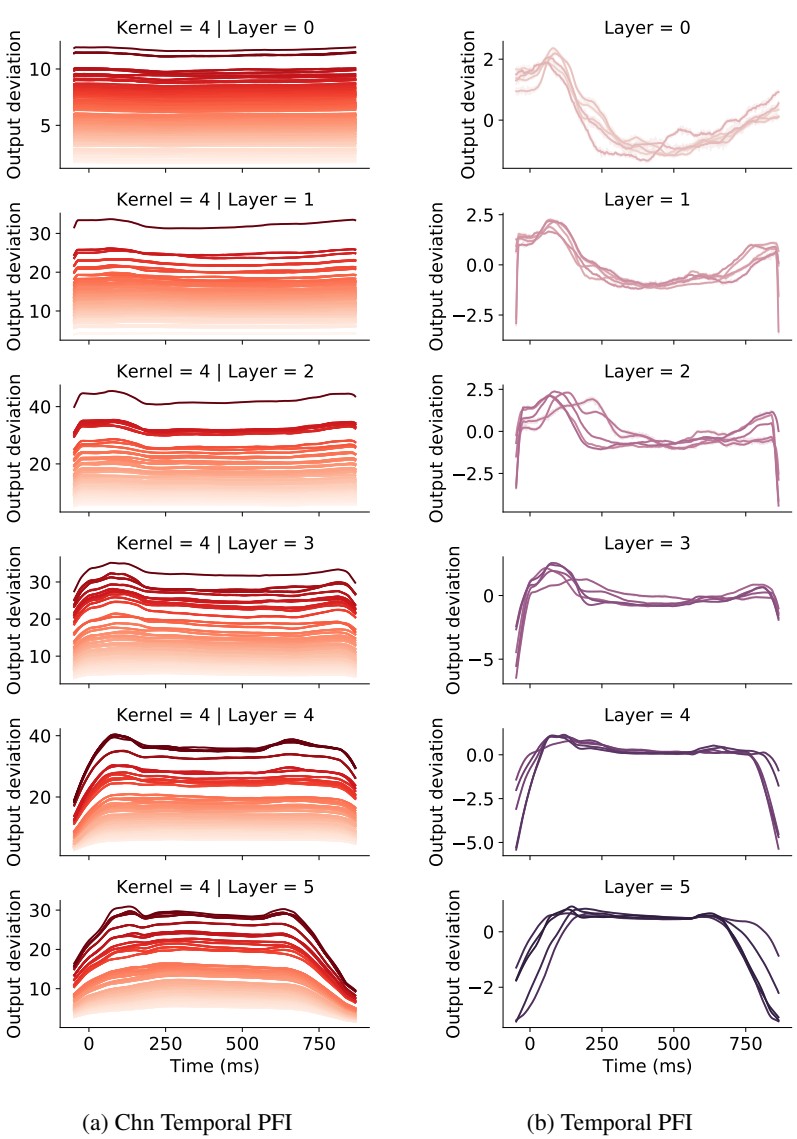

(a) Chn Temporal PFI                    (b) Temporal PFI

Figure 12: Channelwise temporal PFI (a), and temporal PFI (b) across kernels of the `non-linear group-emb` model in 6 layers (rows). For temporal PFI 5 kernels (lines) are plotted together. Channelwise temporal PFI shows the temporal PFI of each channel for Kernel 5. Channel coloring is matched to the corresponding spatial PFI map, and darker reds mean higher output deviation. For temporal PFI output deviation is normalized. The horizontal axis shows time elapsed since image presentation for both temporal PFI types. 95% confidence interval is shown with shading.

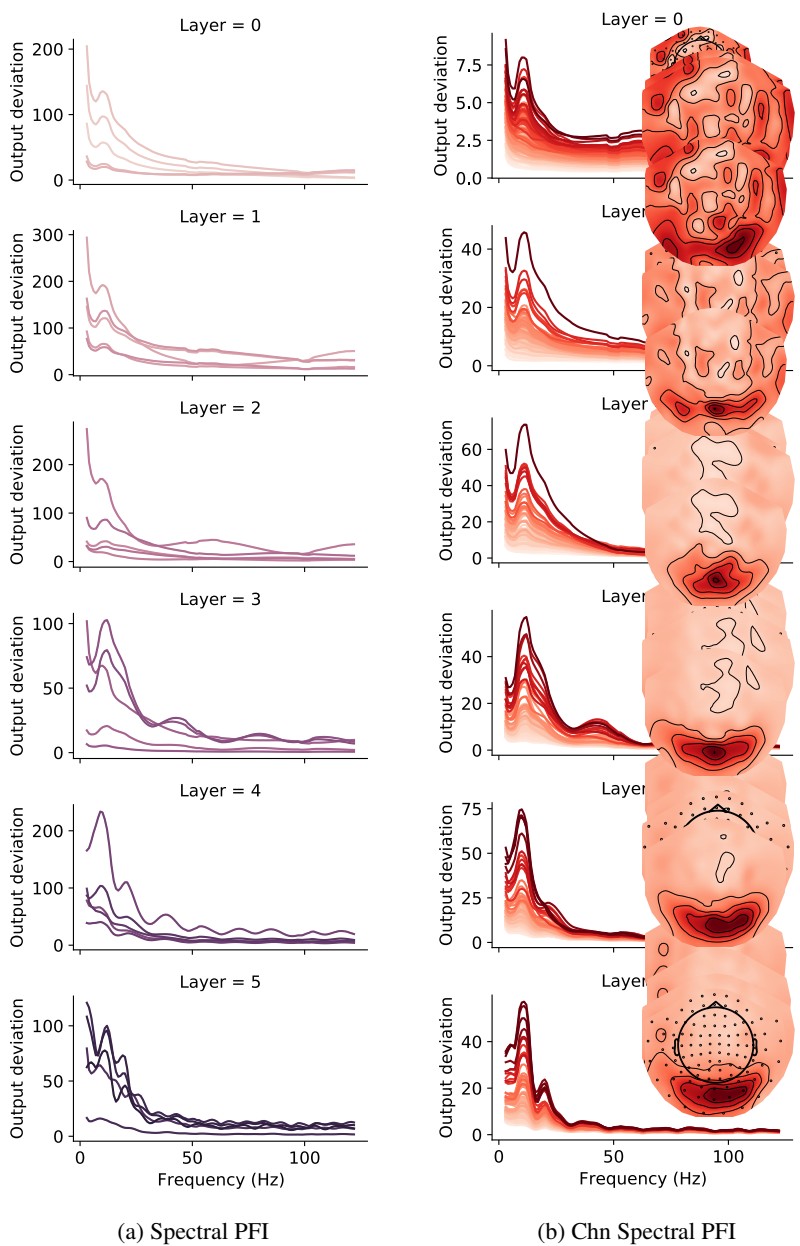

(a) Spectral PFI

(b) Chn Spectral PFI

Figure 13: Frequency sensitivity of kernels via spectral PFI (a), channelwise spectral PFI (b) from 6 layers (rows). Kernels are plotted together (lines) for spectral PFI. Each channelwise spectral PFI plot is for 1 kernel, where lines show the spectral PFI of corresponding channels in the topomap. 95% confidence interval is shown with shading for spectral PFI. Due to small variability across permutations this is barely visible.

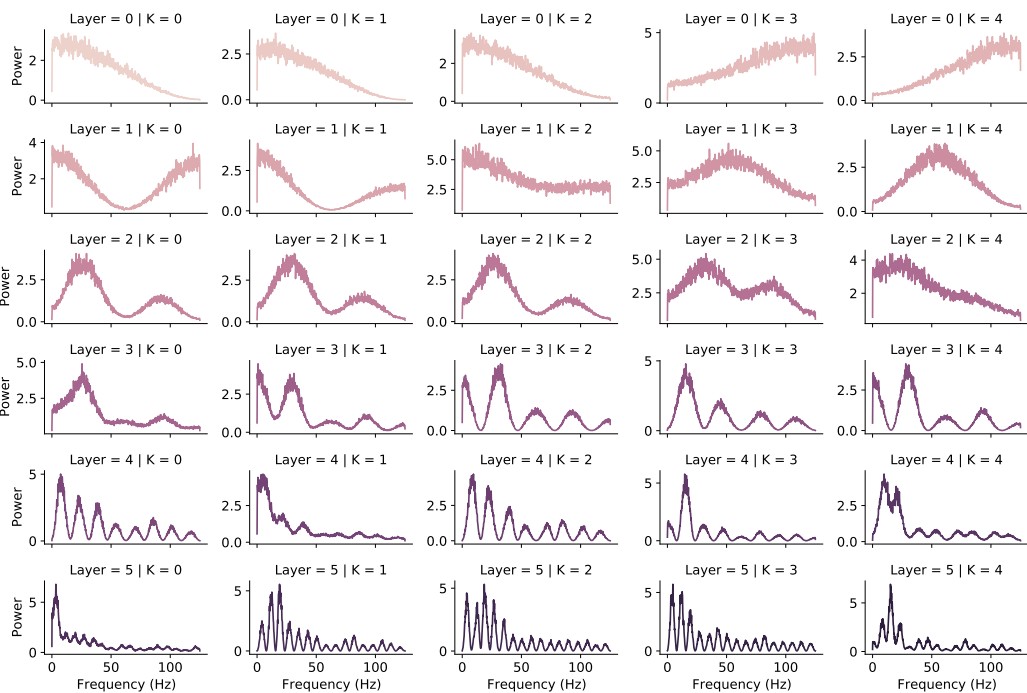

Figure 14: Frequency characteristics of 5 kernels across 6 layers (rows) via kernel FIR analysis. The power spectra is normalized.

