# OpenReview forum: "Group-level Brain Decoding with Deep Learning"
_ICLR.cc/2023/Conference — Submitted to ICLR 2023_

### Official Review · Reviewer_kPfZ · 2022-10-23

**Confidence:** 4
**Correctness:** 2
**Technical Novelty And Significance:** 2
**Empirical Novelty And Significance:** 2
**Recommendation:** 3

**Clarity, Quality, Novelty And Reproducibility:**

Quality: Not very good. The novelty is very limited. Although the experimental results show that the proposed method can significantly improve the performance, the authors only validate their method on one MEG dataset and compare it with weak baselines.
Clarity: The method description is clear. But some detailed descriptions on the experimental results are not very clear.
Originality of the work: not very original.
Reproducibility: The code is publicly available, and the datased is also publicly available. The reproducibility is high.

**Details Of Ethics Concerns:**

No ethics concerns.

**Strength And Weaknesses:**

Strength:
1.This paper builds a GL neural decoding model that can be adapted to different subjects. The key module is the learnable subject embedding, which is simple to apply. In this way, researchers can aggregate multi-subject neural signals to train a GL model and finetune it on the specific target subject. Considering the sample size of neural datasets is always limited, the GL model is meaningful to improve the model performance. Although the model is only validated on an MEG dataset in this paper, I think it can also be applicable to other modalities, such as EEG and fMRI.

Weaknesses:
1.The methodology novelty is very limited. The feature extraction network is similar to the WaveNet. The subject embedding trick and the analyses on the relationship between neural responses and features with PFI have been applied to MEG neural encoding in (Chehab et al. 2021).

2.The paper is a work in progress. The model is only validated on an MEG dataset. It is recommended that the authors perform validation on at least two neural dataset. The results and analyses are not impressive.

3.The LOSO experimental results show that the advantage of the subject embeddings cannot transfer to new subjects. How to leverage transfer learning with the proposed model and learn a useful embedding for the new subject in an unsupervised manner as described in Page 7? The authors only show a concept and don’t conduct any experiment.

4.Many descriptions on the experimental results rather than concrete data in the form of figures and tables. For example, the results of finetuning a naive group model indicated in Page 6 have not been shown in Figure 3. In Section 4.2, the impact of the network layer on accuracy is not shown in corresponding charts. The authors said that the visualization of subject embeddings did not show any clusters and no visualization is shown in the paper or the appendix.

5.Many details are missing. For example, why did the authors change the embedding dimensionality from 10 only to 3 and 14? How about other settings?

6.The improvement of GL model compared with the naïve GL model is significant. However, the naïve group models are too weak to be supportive baseline models. A network module is also proposed in the literature [1] to address the between-subject variability. I suggest that the authors should take a boarder review and compare their model with stronger baselines.

[1] Défossez, A., Caucheteux, C., Rapin, J., Kabeli, O., & King, J. R. (2022). Decoding speech from non-invasive brain recordings. arXiv preprint arXiv:2208.12266.

**Summary Of The Paper:**

This paper proposed a deep learning-based group-level (GL) neural decoding model that can be adapted to different subjects. The key module of the GL model is the learnable subject embeddings. Experiments were conducted on an MEG neural dataset. Experimental results indicated that the proposed GL model can achieve significantly higher classification accuracy than the naive GL modelling. The paper also gained insight into how subject embeddings help the group model and analyzed the spatial and temporal feature importance.
The main contributions of this paper are as follows.
1.The paper builds a GL neural decoding model that can be adapted to different subjects.
2.The paper gains insights into the learned subject embedding.
3.The paper gives analysis of model weights to reveal how meaningful spatio-temporal and spectral information is encoded.

**Summary Of The Review:**

More than an innovative work on representation learning, the work applies an existing method to a new task. I think the quality of the paper is not up to the average quality of the papers accepted by ICLR. Also, the paper is not quite in line with the main scope of ICLR. The article also does a rich analysis from the neuroscience perspective. I suggest the authors to submit their paper to conferences or journals in the field of computational neuroscience after major revisions.

---

> ### Author Response · Authors · 2022-11-15
> **Thank you**
>
> We thank you for taking the time to review our article and for your insightful comments. In the following, we provide responses to each of your concerns.
>
> > The methodology novelty is very limited. The feature extraction network is similar to the WaveNet. The subject embedding trick and the analyses on the relationship between neural responses and features with PFI have been applied to MEG neural encoding in (Chehab et al. 2021).
> * We appreciate that the methodology novelty may be limited, however, our novelties lie in the specific contributions that we make. These are listed in the last paragraph of Section 1. Namely, *“1. We introduce a group level model with subject embeddings, improving on naive group modelling. 2. We provide insight on how non-linearity and subject embedding helps group modelling. 3. Analysing the deep learning-based decoding model provides neuroscientific insights. 4. Analysis of model weights reveals how meaningful spatio-temporal and spectral information is encoded.”* While the subject embedding itself is not a novel idea, we believe our methodology of applying it to MEG decoding is. Furthermore, the finetuning approach presented in section 4 which achieves the best overall performance using the pretrained embedding-aided group model is novel to the best of our knowledge. Our novelty also lies in the way we analyse the inner representations of the model through temporal, spatial, and spectral PFI. Through these analyses, we show neural activity specific to visual information processing.
> * We have now also added the following sentence at the start of section 3.2 making it clear that our Wavenet model is standard, but the application to neural decoding is novel. *“We do not aim to design a new kind of architecture for decoding MEG data, but rather build our model based on previously proven CNN-based architectures for MEG data. While Wavenet has proved to be successful in the audio domain, to the best of our knowledge, there is no prior work applying Wavenet to neural decoding.”*
>
> > The paper is a work in progress. The model is only validated on an MEG dataset. It is recommended that the authors perform validation on at least two neural dataset. The results and analyses are not impressive.
> * Thank you for this suggestion. We have now updated the paper with the following results and discussion. *“We analysed our main findings on another publicly available visual MEG dataset with 92 different images (15 subjects, and 30 trials per image) (Cichy et al., 2014). Linear SL models achieved 35% accuracy, whereas a linear group model without embeddings had 12%, and a nonlinear group model with embeddings had 28%. Thus we can see that our approach behaves similarly on this dataset, improving a lot over the naive group baseline, but not quite achieving the performance of the linear SL models. Finetuning the group model separately on individual subjects achieved 38% accuracy surpassing from-scratch SL models.”*
>
> > The LOSO experimental results show that the advantage of the subject embeddings cannot transfer to new subjects. How to leverage transfer learning with the proposed model and learn a useful embedding for the new subject in an unsupervised manner as described in Page 7? The authors only show a concept and don’t conduct any experiment.
> * Indeed this is a weakness of the approach, however our primary goal as discussed in the introduction was to improve across-subject decoding, and we only conducted LOSO analysis for completeness' sake. We apologise for not making it more clear how our group model can be useful. We have now updated the paper with the following discussion. *“Other than being useful for fine-tuning, our embedding-aided group model can be useful in the case of much larger datasets where we cannot afford to have a separate model for each subject. As demonstrated, neuroscientific investigations can be conducted at the group level on a single group model, instead of many individual subject models (sections 4.4 and 4.5). In neural decoding, group models are widely understood to perform worse than individual models (Guggenmos et al., 2018; Dash et al., 2020). But why is this? By plotting per-subject performance in both kinds of models (see Figure 8 in the Appendix), we see something revealing. While group results are worse on average, some subjects are decoded better in the group model than in their individual models. These generally appear to be the subjects who perform worst on the individual models. This suggests that group models could be successfully used for some subjects if those subjects could be identified. Our results suggest follow-up studies to understand why some subjects performed better or worse.”*

---

> ### Author Response · Authors · 2022-11-15
> **Response part 2**
>
> > Many descriptions on the experimental results rather than concrete data in the form of figures and tables. For example, the results of finetuning a naive group model indicated in Page 6 have not been shown in Figure 3. In Section 4.2, the impact of the network layer on accuracy is not shown in corresponding charts. The authors said that the visualization of subject embeddings did not show any clusters and no visualization is shown in the paper or the appendix.
> * We apologise for not including the fine-tuned naive group model in Figure 3.  We have now rectified this in the updated version of the paper.
> * We apologise for not including the results with different number of layers. We have now updated the paper with the table below showing the effect of the number of layers.
> * We apologise for not including the visualization of the subject embedding clustering, which we have rectified in the updated version of the paper. See below a 2D tsne projection of the embeddings.
>
> [layer results](https://i.ibb.co/42fbmwn/results.png)
>
> [tsne projection](https://i.ibb.co/vPTfBQ6/tsne-emb.png)
>
> > Many details are missing. For example, why did the authors change the embedding dimensionality from 10 only to 3 and 14? How about other settings?
> * We apologise for not discussing this, and we have now added the following sentences to the paper. *“We tried these two settings to see how embedding size in the lower and upper limits influences performance. As an embedding dimensionality of 14 performs the same as 10, we could draw the conclusion that 10 is not a limiting factor. From the much worse result with an embedding dimensionality of 3 we could draw the conclusion that restricting the embedding size too much is not possible.”*
> * We could have evaluated the model with every possible embedding size, for example, to find the lowest embedding size which performs the same as 10, however, this is computationally taxing, and we did not see any usefulness in carrying out such an analysis.
>
> > The improvement of GL model compared with the naïve GL model is significant. However, the naïve group models are too weak to be supportive baseline models. A network module is also proposed in the literature [1] to address the between-subject variability. I suggest that the authors should take a boarder review and compare their model with stronger baselines.
> * In general, we found most related papers are focusing on how to adapt a model trained on one subject to a different subject - we had referred to these in section 2. However, in our work, we wanted to train a single model across a group of subjects (that might later be used for finetuning). We were unable to find works looking into how to improve performance specifically in this way. Thus we think our naive group model is a solid baseline and provides an apples-to-apples setup that allows us to see the effect of subject embeddings in isolation.
> * Please note that as per the last question in the reviewer guidelines, we are not required to compare our results with any paper that came out in the last 4 months (before the submission deadline). In addition the paper in question is a non-peer reviewed article. Even though the referenced paper seems relevant and interesting, we think it is unreasonable to make the request of comparing with an arxiv paper which was uploaded 1 month before the submission deadline.
>
> > More than an innovative work on representation learning, the work applies an existing method to a new task. I think the quality of the paper is not up to the average quality of the papers accepted by ICLR. Also, the paper is not quite in line with the main scope of ICLR. The article also does a rich analysis from the neuroscience perspective. I suggest the authors to submit their paper to conferences or journals in the field of computational neuroscience after major revisions.
> * We would like to note that ICLR is the International Conference on Learning Representations. As such, we analyse and provide insight into a number of different learned representations in our paper, such as the kernels inside the Wavenet Classifier using temporal, spatial, and spectral PFI (section 4.5), and the subject embeddings themselves (section 4.2). Such analyses could be useful for the wider machine learning community as well. As described in lines 4-5 of the first paragraph in section 1, analysing deep learning methods on neuroimaging data is beneficial for the machine learning community, as the small, noisy, high-dimensional real-world datasets test the limits of popular architectures and provide insight into developing new methods. The ICLR website also specifically states that topics of interest include applications in neuroscience.

---

### Official Review · Reviewer_bWS9 · 2022-10-24

**Confidence:** 4
**Correctness:** 3
**Technical Novelty And Significance:** 2
**Empirical Novelty And Significance:** 2
**Recommendation:** 5

**Clarity, Quality, Novelty And Reproducibility:**

- I'm worried about the significance of this work. The presented group-level model performs 6% worse than the subject-level baseline. Fine-tuning on the subject data results in a 5% performance increase, but this greatly narrows the real world applicability of this approach. In the very best case, to see any benefit from this method, one has to pool all their data, train a group level model, and then fine-tune these weights for each subject, essentially doubling the training time and size of the architecture. And even then, the benefit is only a 5% increase.
- And this benefit is not uniform across subjects, indeed for some subjects, the performance slightly decreases (first paragraph of pg. 6).
- The novelty of the approach also concerns me. Compared to the naive group-level approach, the presented approach is essentially identical, but with an extra piece of input data, namely the identity of the subject. Simplicity in itself is not necessarily a bad thing. But I would argue that this paper does not present any new approach, but rather a simple comparison between models: one which has access to a particular bit of information, and one that does not.

**Strength And Weaknesses:**

# Strengths
- The text is well written
- The analysis of MEG signal using trained model weights (section 4.5) shows promise for future neuroscientific study.

# Weaknesses
- The model is not suitable for use in transfer learning. The leave-one-subject-out analysis shows that the naive baseline performs just as well as the proposed model, and that better group performance does not translate to better fine-tuning performance on the held-out subject. The authors rightly note that this is a non-trivial problem, but similar transfer-learning solutions have shown good progress for NLP and vision, and it would greatly strengthen the significance of these results if something analogous could be accomplished for the MEG domain.
- It should also be noted that the group-level architecture is ~2x larger than the subject-level architecture. The authors say that subject-level performance plateaus beyond a certain number of parameters (3 layers), but it would be nice to see this in a table.  Alternatively, the authors could show results for a group-level architecture that is the same size (3 layers) as the subject-level model. As it stands, there remains room to believe that the presented gains over the baseline could be entirely explained by simply using a larger model.
- See below for comments on significance and novelty

**Summary Of The Paper:**

This work is concerned with improving classification of brain activity (MEG). Typically, this would be done by training a model for each subject. But this work presents a strategy for pooling data across multiple subjects (n=15) to produce a "group-level" model. In the naive approach, this can be done by combining data from multiple subjects into one training set. But in this work, they augment each training example with a subject embedding, which is updated during training. They find that a group-level model produces predictions which have a 6% disadvantage to the subject-level models, but a 5% advantage after fine-tuning. The paper also includes an analysis of the model's trained weights and what they reveal about visual processing in the brain.

**Summary Of The Review:**

In the work's current state, I vote to reject. The approach as it stands cannot be used for transfer-learning, which could be one of the main benefits of a group-level model, if accomplished. In its current form, the proposed method doubles the training time and number of weights in order to achieve a 5% increase over the baseline subject-level model.

---

> ### Author Response · Authors · 2022-11-15
> **Thank you**
>
> We thank you for taking the time to review our article and for your insightful comments. In the following, we provide responses to each of your concerns.
>
> > The model is not suitable for use in transfer learning. The leave-one-subject-out analysis shows that the naive baseline performs just as well as the proposed model, and that better group performance does not translate to better fine-tuning performance on the held-out subject. The authors rightly note that this is a non-trivial problem, but similar transfer-learning solutions have shown good progress for NLP and vision, and it would greatly strengthen the significance of these results if something analogous could be accomplished for the MEG domain.
> * Indeed this is a weakness of the approach, however our primary goal as discussed in the introduction was to improve across-subject decoding, and we only conducted LOSO analysis for completeness' sake. We apologise for not making it more clear how our group model can be useful. We have now updated the paper with the following discussion. *“Other than being useful for fine-tuning, our embedding-aided group model can be useful in the case of much larger datasets where we cannot afford to have a separate model for each subject. As demonstrated, neuroscientific investigations can be conducted at the group level on a single group model, instead of many individual subject models (sections 4.4 and 4.5). In neural decoding, group models are widely understood to perform worse than individual models (Guggenmos et al., 2018; Dash et al., 2020). But why is this? By plotting per-subject performance in both kinds of models (see Figure 8 in the Appendix), we see something revealing. While group results are worse on average, some subjects are decoded better in the group model than in their individual models. These generally appear to be the subjects who perform worst on the individual models. This suggests that group models could be successfully used for some subjects if those subjects could be identified. Our results suggest follow-up studies to understand why some subjects performed better or worse.”*
>
> > It should also be noted that the group-level architecture is ~2x larger than the subject-level architecture. The authors say that subject-level performance plateaus beyond a certain number of parameters (3 layers), but it would be nice to see this in a table. Alternatively, the authors could show results for a group-level architecture that is the same size (3 layers) as the subject-level model. As it stands, there remains room to believe that the presented gains over the baseline could be entirely explained by simply using a larger model.
> * We apologise for not including the actual accuracy values in the text in the original submission. We have now updated section 4.2 with the following discussion. *“For non-linear SL models using 6 layers results in an 14% drop in accuracy compared to using 3 layers. For linear SL models, the drop in accuracy is about 4%. In the case of GL models using 3 layers instead of 6 results in a 16% drop in accuracy. To be clear, because of how we perform the downsampling after the convolutional layers (described in section 3.2), using fewer convolutional layers actually increases the overall parameter count because the fully-connected block has to be enlarged. Thus, the group model (with 6 conv layers), is about 2.5x smaller than the SL models (with 3 conv layers).”* Thus, the results could not be explained by using a larger model, as the group model is actually smaller than the subject-level model in terms of parameter count.

---

> > ### Comment · Reviewer_bWS9 · 2022-11-28
> > **Reply to authors**
> >
> > I thank the authors for taking the time to reply and especially for clarifying the size of the SL and GL models. But my reasons for the original score still stand. (1) The significance of the results is more on the marginal side. The model shows small and situational improvement over the baselines. And (2) the technical contribution of this work seems rather limited for the ML community and the bulk of the analyses seem like they would have a better fit at a neuroscience venue.

---

> ### Author Response · Authors · 2022-11-15
> **Response part 2**
>
> > I'm worried about the significance of this work. The presented group-level model performs 6% worse than the subject-level baseline. Fine-tuning on the subject data results in a 5% performance increase, but this greatly narrows the real world applicability of this approach. In the very best case, to see any benefit from this method, one has to pool all their data, train a group level model, and then fine-tune these weights for each subject, essentially doubling the training time and size of the architecture. And even then, the benefit is only a 5% increase.
> * As mentioned before, the group model is actually smaller than the subject-level model. Considering these aspects, the 5% increase is entirely reasonable from an applicability point of view. Most machine learning papers often sacrifice large amounts of computing and memory to achieve a few percentage increases. Please note, however, in this work our aim was not to find the smallest working models, but rather to allow both subject-level and group-level models to be as big as required (with a high amount of dropout) to achieve the best performance in both scenarios, respectively. This is to show a fair comparison between the two. If one’s goal is real-world application where compute and memory restrictions are in place, then model size could probably be greatly reduced in clever ways (e.g. knowledge distillation, or smaller models with less dropout). Thus, we do not think that any conclusions regarding compute and memory-restricted applications of our results can be made, unfortunately.
> * It is true that the group model is 6% worse than the subject-level model, however, it is a huge improvement compared to the baseline naive group model (24% increase). Please refer to our response to a previous concern detailing how we updated the paper with a discussion on how our group model can be useful.
>
> > And this benefit is not uniform across subjects, indeed for some subjects, the performance slightly decreases (first paragraph of pg. 6).
> * It is indeed true that performance decreased for 2 out of 15 subjects. However, we think this reveals interesting information about between-subject variability, and warrants follow-up studies to understand, as mentioned in our previous responses.
>
> > The novelty of the approach also concerns me. Compared to the naive group-level approach, the presented approach is essentially identical, but with an extra piece of input data, namely the identity of the subject. Simplicity in itself is not necessarily a bad thing. But I would argue that this paper does not present any new approach, but rather a simple comparison between models: one which has access to a particular bit of information, and one that does not.
> * We have listed our contributions in the last paragraph of Section 1. We believe that a comparison between two models, where the difference between the models impacts the objectives of the approach is clearly of interest. This is a common approach taken in such papers. While the subject embedding itself is not a novel idea, we believe our methodology of applying it to MEG decoding is. Furthermore, the finetuning approach presented in section 4 which achieves the best overall performance using the pretrained embedding-aided group model is novel to the best of our knowledge. Our novelty also lies in the way we analyse the inner representations of the model through temporal, spatial, and spectral PFI. Through these analyses, we show neural activity specific to visual information processing.

---

### Official Review · Reviewer_NEjD · 2022-10-25

**Confidence:** 5
**Correctness:** 3
**Technical Novelty And Significance:** 2
**Empirical Novelty And Significance:** 3
**Recommendation:** 5

**Clarity, Quality, Novelty And Reproducibility:**

The dataset is freely available, but the GitHub code would probably be available after manuscript acceptance, which is a problem for the reviewer.
The current ML model and MEG preprocessing descriptions are too vague for reproducibility. The vague description makes the WaveNet application appear standard and without significant novelty.

**Strength And Weaknesses:**

Strength: A relatively standard WaveNet application to MEG is unfortunately out of scope for ICLR.

Weaknesses: Missing technical details. The authors include very vague technical descriptions of MEG preprocessing as follows:
"Raw data is bandpass filtered between 0.1 and 125 Hz, and line noise is removed with notch filters. After downsampling to 250 Hz, 1.024-second epochs are extracted, starting 100 ms before stimulus presentation. This resulted in 306 x 256-dimensional trials (channels x timesteps) from the 306 MEG sensors. Whitening is used to remove covariance between channels for SL models, whereas for GL models, standardization is performed per channel. We do multiclass decoding, predicting a separate probability for each of the 118 classes (images)."
What kind of filters (FIR, IIR, etc.) were used? What downsampling and whitening procedures were applied (the reviewer had no access to the code)? Why 118 classes (that would be a rather revolutionary BCI)?

Similar problems continue with the WaveNet model's vague description.


**Summary Of The Paper:**

The authors present an interesting discussion and MEG classification results in SL vs. GL settings. The submission is not a good match for ICLR, and it would interest the applied neuroscience community. The technical contribution of a relatively standard/off-the-shelf WaveNet to MEG is the major weakness of the paper. Technical details on the reproduction of the model (see details in a section below) are missing, and a promise in the abstract, "All code is available on GitHub," lands an anonymized link (a GitHub search for the code would result probably in the authors' identification).

**Summary Of The Review:**

The manuscript is not a good match for the ICLR. Little ML novelty and extended results discussion from a neuroscientific application point of view would contribute to the SfN poster but not the ICLR.

---

> ### Author Response · Authors · 2022-11-15
> **Thank you**
>
> We thank you for taking the time to review our article and for your insightful comments. In the following, we provide responses to each of your concerns.
>
> > Technical details on the reproduction of the model (see details in a section below) are missing, and a promise in the abstract, "All code is available on GitHub," lands an anonymized link
> * Please note that we had (and have) attached all code as supplementary material, which to our knowledge is the standard way to include code as part of a submission to ICLR.
>
> > What kind of filters (FIR, IIR, etc.) were used? What downsampling and whitening procedures were applied (the reviewer had no access to the code)? Why 118 classes (that would be a rather revolutionary BCI)?
> * We apologise for not including further details, however, we did include technical details regarding preprocessing at an appropriate level of detail in the main body of the paper in Section 3.1. Further implementation details can be found in the code, which is attached as supplementary material. Note that we used FIR filters and a simple decimation for downsampling. For whitening, we just transformed the data using PCA and retained all components. Since the dataset we used contains 118 different images as stimuli, it naturally followed to use 118 classes for decoding. We would also like to clarify that our work is not intended to be related to any kind of BCI application. We have now made this clear by adding the following sentences to the introduction (in the updated version of the paper). *“Our aim in this paper is to improve across-subject decoding of MEG data by using a group model that generalizes across subjects. To be clear this objective and the datasets we use are not directly related to any kind of direct BCI application.”*
>
> > Similar problems continue with the WaveNet model's vague description.
> * As with the preprocessing details we feel that we did include all relevant technical details at an appropriate level of detail for the main body of the paper. The exact implementation can be found in our code which is attached as supplementary material.
>
> > The dataset is freely available, but the GitHub code would probably be available after manuscript acceptance, which is a problem for the reviewer. The current ML model and MEG preprocessing descriptions are too vague for reproducibility. The vague description makes the WaveNet application appear standard and without significant novelty.
> * As described in an earlier response, all our results can be easily reproduced by running the code that was, and is, in the supplementary materials. Our contributions are listed in the last paragraph of Section 1. We have now also added the following sentence at the start of section 3.2 making it clear that our Wavenet model is standard, but the application to neural decoding is novel. *“We do not aim to design a new kind of architecture for decoding MEG data, but rather build our model based on previously proven CNN-based architectures for MEG data. While Wavenet has proved to be successful in the audio domain, to the best of our knowledge, there is no prior work applying Wavenet to neural decoding.”*
>
> > The manuscript is not a good match for the ICLR. Little ML novelty and extended results discussion from a neuroscientific application point of view would contribute to the SfN poster but not the ICLR.
> * We would like to note that ICLR is the International Conference on Learning Representations. As such, we analyse and provide insight into a number of different learned representations in our paper, such as the kernels inside the Wavenet Classifier using temporal, spatial, and spectral PFI (section 4.5), and the subject embeddings themselves (section 4.2). Such analyses could be useful for the wider machine learning community as well. As described in lines 4-5 of the first paragraph in section 1, analysing deep learning methods on neuroimaging data is beneficial for the machine learning community, as the small, noisy, high-dimensional real-world datasets test the limits of popular architectures and provide insight into developing new methods. The ICLR website also specifically states that topics of interest include applications in neuroscience.
> * While the subject embedding itself is not a novel idea, we believe our methodology of applying it to MEG decoding is. Furthermore, the finetuning approach presented in section 4 which achieves the best overall performance using the pretrained embedding-aided group model is novel to the best of our knowledge.

---

### Official Review · Reviewer_MQXD · 2022-10-25

**Confidence:** 3
**Correctness:** 3
**Technical Novelty And Significance:** 3
**Empirical Novelty And Significance:** 3
**Recommendation:** 6

**Clarity, Quality, Novelty And Reproducibility:**

The manuscript is fairly clearly written, seems novel to the best of my knowledge and code is available.

**Strength And Weaknesses:**

**Update**
Due to the including of the reference, the clarifications regarding number of layers and the additional gradient-based analyses, have increased my score.

**Pre-rebuttal**
The manuscript is quite detailed and evaluates many aspects of a clearly described idea.

It is valuable that the work looks at various settings like linear/nonlinear with/without embedding and also including subject-level finetuning. Also the additional analysis of the trained models is interesting.

Some open questions for me:
“For SL modelling, the Wavenet Classifier contains 3 convolutional layers, whereas for group modelling it has 6, further motivated in Section 4. “

I could not quite find the part where this is written maybe I missed it? In any case, would be nice to also evaluate the 6-layer classifier on subject-level decoding, as it may also extract different frequencies with its deeper structure. So these results would be an important addition, also to disentangle effects of larger model and effect of subject embedding etc.

In general, it would be good to evaluate further published EEG deep learning models/pipelines on this task to know how the reported accuracies compare to the literature. Also, I assume this dataset has been decoded before, how where accuracies in the literature?

I was also missing at least one reference on transfer learning for neurophysiological recordings,
https://iopscience.iop.org/article/10.1088/1741-2552/abb7a7
Also this or other works would provide interesting baselines for the method presented in this manuscript.


Regarding the PFI analysis, it would be nice to additionally perform a computationally simpler gradient-based analysis, note that both fourier transform and inverse fourier transform are differentiable and hence one can also compute gradients on fourier coefficients. It would be good to see if these analysis agree or in how far they disagree.


**Summary Of The Paper:**

This manuscript analyzes the use of subject embeddings to enhance deep learning models trained to decode MEG recordings of several subjects. Concretely, the work analyzes decoding MEG data to classify which of 118 different images 15 different subjects looked at. They use a convolutional neural network using dilated convolutions inspired by WaveNet. and evaluate using a vanilla model without subject embeddings as well as a larger model that also uses subject embeddings. They report that using the subject embeddings improves accuracies and when combined with finetuning on an individual subject can even slightly outperform subject-level models. Furthermore, analysis of the trained models partly shows plausible results.

**Summary Of The Review:**

Overall, this seems a nice manuscript with some additional analyses needed in my view.

---

> ### Author Response · Authors · 2022-11-15
> **Thank you**
>
> We thank you for taking the time to review our article and for your insightful comments. In the following, we provide responses to each of your concerns.
>
> > “For SL modelling, the Wavenet Classifier contains 3 convolutional layers, whereas for group modelling it has 6, further motivated in Section 4. “ I could not quite find the part where this is written maybe I missed it? In any case, would be nice to also evaluate the 6-layer classifier on subject-level decoding, as it may also extract different frequencies with its deeper structure. So these results would be an important addition, also to disentangle effects of larger model and effect of subject embedding etc
> * We have discussed the effect of the number of layers on performance in lines 5-10 in section 4.2. As written, we found SL models to be considerably worse by increasing the number of layers and GL models to be considerably better. We apologise for not including the actual accuracy values in the text in the original submission. We have now updated section 4.2 with the following discussion. *“For non-linear SL models using 6 layers results in an 14% drop in accuracy compared to using 3 layers. For linear SL models, the drop in accuracy is about 4%. In the case of GL models using 3 layers instead of 6 results in a 16% drop in accuracy. To be clear, because of how we perform the downsampling after the convolutional layers (described in section 3.2), using fewer convolutional layers actually increases the overall parameter count because the fully-connected block has to be enlarged. Thus, the group model (with 6 conv layers), is about 2.5x smaller than the SL models (with 3 conv layers).”*
>
> > In general, it would be good to evaluate further published EEG deep learning models/pipelines on this task to know how the reported accuracies compare to the literature. Also, I assume this dataset has been decoded before, how where accuracies in the literature?
> * To the best of our knowledge, no previous multiclass decoding results exist for this dataset. While we appreciate the importance of comparison with other published models, our main aim here was not to improve classification accuracy on this dataset in general, but to bridge the gap between subject and group-level models. To this end, we believe the  results that we present in the paper are useful and allow us to see the effect of subject embeddings in isolation. There are a plethora of different architectures used for EEG or MEG decoding in the literature, and our CNN-based model can be considered relatively standard.
>
> > I was also missing at least one reference on transfer learning for neurophysiological recordings. Also this or other works would provide interesting baselines for the method presented in this manuscript.
> * Thank you for this reference which we have now included in the updated version of the paper with the following discussion: *“The euclidean alignment is very similar to a spatial whitening of the data. We have tried this in conjunction with our group model, however, we found this to lower performance, and thus opted for a simpler channel-wise standardization. The mixup regularisation mentioned in Kostas at el. is entirely complementary to our approach and can be used in conjunction. This is a general regularization/data augmentation technique and does not specifically deal with inter-subject variability.”*
> * In general, we found most related papers are focusing on how to adapt a model trained on one subject to a different subject - we had referred to these in section 2. However, in our work, we wanted to train a single model across a group of subjects (that might later be used for finetuning). We were unable to find works looking into how to improve performance specifically in this way.
>
> > Regarding the PFI analysis, it would be nice to additionally perform a computationally simpler gradient-based analysis, note that both fourier transform and inverse fourier transform are differentiable and hence one can also compute gradients on fourier coefficients. It would be good to see if these analysis agree
> * Thank you for this useful analysis recommendation. We ran the gradient-based analysis and attach the figures below. We have now updated the paper with the following discussion and the figures below. *“We compared the accuracy-based PFI analysis with a standard gradient-based analysis where a saliency map is obtained by backpropagating to randomly initialized inputs. We smoothed the temporal profile with the same window size as for the PFI analysis. Temporally we can see that the agreement between the two methods is high, with peaks aligning very well (less than 10ms difference). Spatially the two methods do show some differences, but overall gradient analysis still points to the most important information being in the visual cortex.”*
>
> [spatial gradients](https://i.ibb.co/qk0HFqZ/spatial-gradients.png)
>
> [temporal gradients](https://i.ibb.co/FzmP8QL/temporal-gradient.png)

---

> > ### Comment · Reviewer_MQXD · 2022-11-23
> > **Thanks; increased score**
> >
> > Thanks for  including the reference, the clarifications regarding number of layers and the additional gradient-based analyses, I have increased my score.
> > I still think including some analyses on existing model architectures would be increase the value of any future submission.

---

### Decision · Program_Chairs · 2023-01-20

**Decision:**

Reject

**Justification For Why Not Higher Score:**

Even the most enthusiastic reviewers were not particularly convinced. The method is of limited interest for ICLR; it's unclear how this community can engage with the work. The core idea behind the method is well-known and the prospects for building on the work are limited.

**Justification For Why Not Lower Score:**

N/A

**Metareview: Summary, Strengths And Weaknesses:**

Summary: Improving decoding of brain activity from MEG by aggregating data across subjects while tagging the training data with a subject-specific embedding.

Strengths: The method may be useful for increasing decoding performance.

Weaknesses: The method is of limited interest and this is reflected in the scores. It's unclear what the rest of the community has to learn from the method or add to it. The general idea of subject/task embeddings is not new and the novelty of the application is limited. A different venue may well receive the work with more enthusiasm.

**Summary Of Ac-Reviewer Meeting:**

N/A